# In-silico-assisted derivatization of triarylboranes for the catalytic reductive functionalization of aniline-derived amino acids and peptides with $H_2$

Yusei Hisata[1], Takashi Washio [2], Shinobu Takizawa [3], Sensuke Ogoshi [1] & Yoichi Hoshimoto [1,4] ✉

Cheminformatics-based machine learning (ML) has been employed to determine optimal reaction conditions, including catalyst structures, in the field of synthetic chemistry. However, such ML-focused strategies have remained largely unexplored in the context of catalytic molecular transformations using Lewis-acidic main-group elements, probably due to the absence of a candidate library and effective guidelines (parameters) for the prediction of the activity of main-group elements. Here, the construction of a triarylborane library and its application to an ML-assisted approach for the catalytic reductive alkylation of aniline-derived amino acids and C-terminal-protected peptides with aldehydes and $H_2$ is reported. A combined theoretical and experimental approach identified the optimal borane, i.e., $B(2,3,5,6\text{-}Cl_4\text{-}C_6H)(2,6\text{-}F_2\text{-}3,5\text{-}(CF_3)_2\text{-}C_6H)_2$, which exhibits remarkable functional-group compatibility toward aniline derivatives in the presence of 4-methyltetrahydropyran. The present catalytic system generates $H_2O$ as the sole byproduct.

Catalysis is a fact of our daily lives. A wide variety of important commercial chemical substances are currently produced on both the fine and bulk scales in the presence of molecular catalysts that have been optimized based on specific factors such as efficiency, toxicity, cost, or a combination thereof. Recent advancements in cheminformatics-based machine learning (ML) offer chemists a way to bypass traditional Edisonian empiricism and develop more efficient approaches to optimizing catalysts[1–4]. Several groups have reported successful demonstrations of ML-driven optimizations of homogeneous catalysts such as phosphoric acids and Lewis-basic ligands for metal-based catalysts involving phosphines, *N*-heterocyclic carbenes, and nitrogen-based ligands[5–18].

Recent progress in frustrated Lewis pairs (FLPs)[19,20] has expanded the practical and sustainable application of main-group catalysis, e.g.,

enabling the hydrogenation of unsaturated molecules without toxic/precious metals[21–26]. In this context, the main-group-catalyzed reductive alkylation of amines with carbonyl compounds and $H_2$ via the generation of FLP species has been widely accepted as a waste-minimizing process that generates valuable *N*-alkylated amines, whereby $H_2O$ is the only by-product[23,27–31]. Our group[30] and that of Soós[28] have independently shown that triarylboranes effectively catalyze the reductive alkylation of a variety of amines with aldehydes in the presence of $H_2$. Moreover, we have demonstrated that an FLP that consists of Soós' borane, i.e., $B(2,6\text{-}Cl_2\text{-}C_6H_3)(2,3,5,6\text{-}F_4\text{-}C_6H)_2$ ($\mathbf{B^{1a}}$)[27], and tetrahydrofuran (THF) exhibits good functional-group tolerance for aniline derivatives including halogens, hydroxyl, and amide groups (Fig. 1A)[30]. Based on mechanistic studies, we proposed dual catalysis of $\mathbf{B^{1a}}$ in the formation of imine intermediates and the hydrogenation of

[1]Department of Applied Chemistry, Faculty of Engineering, Osaka University, Suita, Osaka 565-0871, Japan. [2]Department of Reasoning for Intelligence and Artificial Intelligence Research Center, SANKEN, Osaka University, Ibaraki, Osaka 567-0047, Japan. [3]Department of Synthetic Organic Chemistry and Artificial Intelligence Research Center, SANKEN, Osaka University, Ibaraki, Osaka 567-0047, Japan. [4]Division of Applied Chemistry, Center for Future Innovation (CFi), Faculty of Engineering, Osaka University, Suita, Osaka 565-0871, Japan. ✉e-mail: hoshimoto@chem.eng.osaka-u.ac.jp

**Fig. 1 | Context of this work. A** Schematic representation of the concept of this study. **B** Proposed dual catalysis of triarylborane in the catalytic reductive alkylation of amines with aldehydes; $B$ = triarylborane; **LB** = THF or MTHP.

the imines via the generation of the FLP species (Fig. 1B). However, the direct reductive alkylation of amino acids with $H_2$ has remained challenging, and such reactions have proceeded in only low-to-moderate yields even under forcing conditions. In terms of toxicity, the solvent THF, which also acts here as a Lewis base to generate FLPs with boranes (Fig. 1B), should be replaced with a less hazardous chemical[32,33]. Given the central role of the reductive alkylation of amines using carbonyl compounds in the synthesis of, e.g., pharmaceuticals, bio-active molecules, and agrochemicals[34–36], the development of a straightforward and greener protocol for derivatizing amino acids and peptides would be worthwhile.

To this end, we envisioned an ML-assisted approach to identify a suitable triarylborane that is able to efficiently catalyze the reductive alkylation of amino acids and peptides with $H_2$ through the construction of an in-silico library that includes a variety of triarylboranes. The synthesis of triarylboranes with unknown substitution patterns is typically a laborious and time-consuming process that often requires several weeks or even months for optimization. However, once the optimal procedures have been obtained, the optimized conditions can often be extrapolated, which is much faster. Therefore, using an ML-

assisted approach to streamlining the selection of triarylborane candidates has the potential to significantly accelerate the optimization process and therefore the entire research process. Moreover, through the construction of this in-silico library, we aimed to contribute to the structural diversification of triarylboranes beyond the archetypical $B(C_6F_5)_3$[37–40], which should expand the utility of this compound class in catalysis, materials science, and other areas. It should be noted that Dyson and Corminboeuf et al. have recently demonstrated the hydrogenation of $CO_2$ to yield a formate salt [DBU–H][H–COO], which was catalyzed by an FLP consisting of tris(*p*-bromo)tridurylborane and 1,8-diazabicyclo[5.4.0]undec-7-ene (DBU)[41]. They identified this combination of borane and DBU using a cheminformatics-assisted approach that profiled the theoretically predicted catalyst activity based on the intrinsic acidity and basicity of the Lewis components.

Herein, we report the construction of an in-silico library with 54 triarylboranes, which was used for the ML-assisted identification of the optimal borane for the catalytic reductive alkylation of aniline-derived amino acids and peptides with aldehydes and $H_2$ (Fig. 1A). We also explored the functional-group compatibility of the present system using the functional group evaluation (FGE) kit recently proposed by

Morimoto and Oshima et al.[42], which is based on the concept of robustness screening that has been proposed by Glorius et al.[43,44].

## Results and discussion

### Theoretical and experimental variables collection

We started our investigation with the construction of an in-silico library of triarylboranes using the strategy described below. Generally, we explored triarylboranes that seemed to be synthetically accessible using common procedures[37,40]. Optimization of the gas-phase structures of the 54 boranes shown in Fig. 2A was accomplished using DFT calculations at the $\omega$B97X-D/6-311+G(d,p)//$\omega$B97X-D/6-31G(d,p) level. The 53 explored heteroleptic boranes $\mathbf{B}^{xy}$ ($x = \mathbf{1}$–$\mathbf{6}$, $y = \mathbf{a}$–$\mathbf{w}$) include two 2,6-F$_2$-3,5-R$_2$-C$_6$H groups, and their core structures are classified as $\mathbf{B^1}$–$\mathbf{B^6}$ depending on the R groups, i.e., $\mathbf{B}^x$(R) = $\mathbf{B^1}$(F), $\mathbf{B^2}$(Cl), $\mathbf{B^3}$(Br), $\mathbf{B^4}$(CF$_3$), and $\mathbf{B^5}$(H), whereas $\mathbf{B^6}$ includes 2,6-F$_2$-3-Cl-C$_6$H$_2$ groups. With these core structures, we combined 23 aryl groups ($\mathbf{a}$–$\mathbf{w}$), and completed the construction of the borane library with the addition of B(C$_6$F$_5$)$_3$. It should be noted that boranes $\mathbf{B}^{xa}$ ($x = \mathbf{1}$–$\mathbf{3}$, $\mathbf{5}$, $\mathbf{6}$), $\mathbf{B}^{1v}$, and $\mathbf{B}^{1w}$ were known before the construction of this library[27,30,45–47], and their reactivity in the hydrogenation of unsaturated molecules inspired us when designing the library molecules. In fact, $\mathbf{B}^{1a}$ has previously been employed for the catalytic reductive alkylation of aniline derivatives to generate active FLP species with THF[30], and thus, in-silico derivatization of $\mathbf{B}^{1a}$ was carried out via substitution of the *meta* and/or *para* H atoms in the 2,6-Cl$_2$-C$_6$H$_3$ group with Cl, Br, CF$_3$, OMe, OCF$_3$ or C$_6$F$_5$ groups. We further conducted extensive in-silico derivatization of $\mathbf{B}^{2a}$ and $\mathbf{B}^{3a}$, given that these boranes exhibit far superior catalytic activity than $\mathbf{B}^{1a}$, $\mathbf{B}^{6a}$, and B(C$_6$F$_5$)$_3$ in the hydrogenation of *N*-heteroaromatics using a gaseous mixture of H$_2$/CO/CO$_2$/CH$_4$[45]. In these cases, we envisioned that the modulation of the intrinsic Lewis acidity of the triarylboranes, i.e., the energy levels of the LUMO, which includes the p orbital on the boron center[48], as well as the remote back-strain[49] that would influence the stability of the four-coordinated tetrahedral Lewis base–borane adducts. The introduction of 2,6-Br$_2$-C$_6$H$_3$ ($\mathbf{j}$) and its derivatives ($\mathbf{k}$–$\mathbf{r}$) into the $\mathbf{B^2}$ core was also explored, as we expected an increase in the front strain that influences the accessibility of the Lewis bases to the boron centers[37].

We subsequently obtained the theoretical parameters. We thought that the use of structural parameters obtained from the gas-phase optimization of $\mathbf{B}^{xy}$ would not play a critical role in predicting the reactivity of the triarylboranes under the chosen conditions, given that no substantial differences were observed among them (Supplementary Table 4). Thus, we obtained the following energetic parameters, calculated at the $\omega$B97X-D/6-311+G(d,p)//$\omega$B97X-D/6-31G(d,p) level: (i) the energy levels of the LUMOs [eV], which include the p orbitals on the boron atoms, (ii) the energy barriers ($\Delta G_H^{\ddagger}$) [kcal mol$^{-1}$] for the heterolytic cleavage of H$_2$ with the combination of $\mathbf{B}^{xy}$ and THF, and (iii) the relative Gibbs energy values ($\Delta G_w^{\circ}$) [kcal mol$^{-1}$] for the formation of the H$_2$O–$\mathbf{B}^{xy}$ adducts with respect to [H$_2$O + $\mathbf{B}^{xy}$]. These theoretical values are shown in Fig. 3A for selected boranes. For parameter (ii), we have previously proposed that the heterolytic cleavage of H$_2$ by FLPs should be involved in the rate-determining event of the $\mathbf{B}^{1a}$-catalyzed reductive alkylation of amines (Path I in Fig. 1B)[30]. For parameter (iii), H$_2$O could be a potential quencher of the triarylborane catalysts via the formation of adducts (Path II in Fig. 1B) followed by proto-deboronation[50]. In this context, we envisioned that boranes $\mathbf{B}^{xy}$ that exhibit larger $\Delta G_w^{\circ}$ and smaller $\Delta G_H^{\ddagger}$ values should show superior performance as Lewis acids for the generation of more active FLPs with ethereal components. It should also be mentioned here that we theoretically optimized a structure that included a sole imaginary

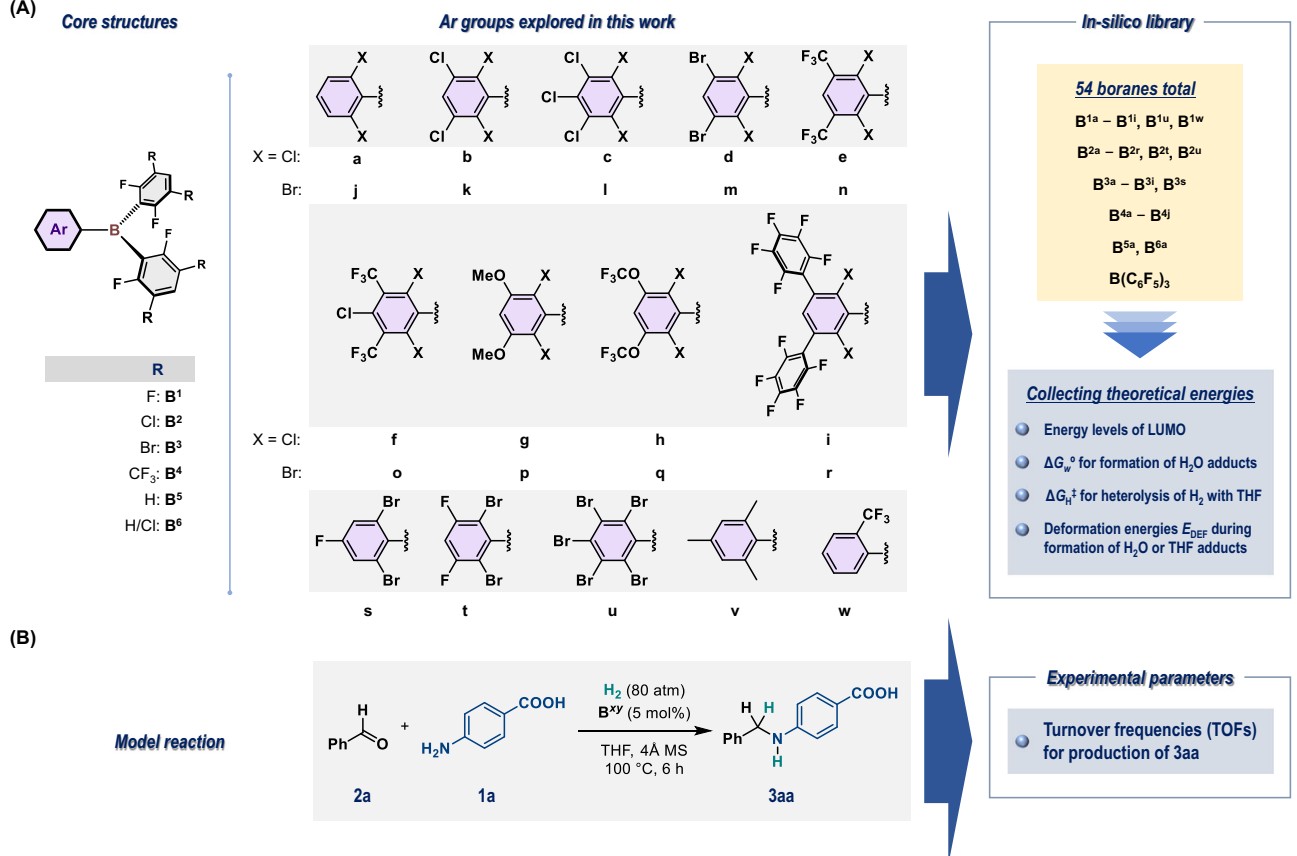

**Fig. 2 | Experiment designs. A** The in-silico library of triarylboranes explored in this work. The structure of $\mathbf{B^6}$ includes 2,6-F$_2$-3-Cl-C$_6$H$_2$ groups. **B** The model reaction for obtaining the experimental parameters (turnover frequencies per hour) used in this work.

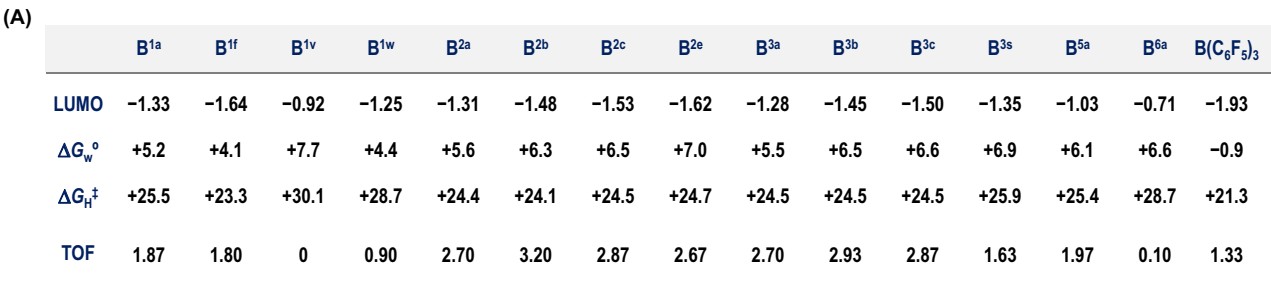

**(A)**

| | B¹ᵃ | B¹ᶠ | B¹ᵛ | B¹ʷ | B²ᵃ | B²ᵇ | B²ᶜ | B²ᵉ | B³ᵃ | B³ᵇ | B³ᶜ | B³ˢ | B⁵ᵃ | B⁶ᵃ | B(C₆F₅)₃ |
|---|---|---|---|---|---|---|---|---|---|---|---|---|---|---|---|
| LUMO | −1.33 | −1.64 | −0.92 | −1.25 | −1.31 | −1.48 | −1.53 | −1.62 | −1.28 | −1.45 | −1.50 | −1.35 | −1.03 | −0.71 | −1.93 |
| $\Delta G_w^\circ$ | +5.2 | +4.1 | +7.7 | +4.4 | +5.6 | +6.3 | +6.5 | +7.0 | +5.5 | +6.5 | +6.6 | +6.9 | +6.1 | +6.6 | −0.9 |
| $\Delta G_H^\ddagger$ | +25.5 | +23.3 | +30.1 | +28.7 | +24.4 | +24.1 | +24.5 | +24.7 | +24.5 | +24.5 | +24.5 | +25.9 | +25.4 | +28.7 | +21.3 |
| TOF | 1.87 | 1.80 | 0 | 0.90 | 2.70 | 3.20 | 2.87 | 2.67 | 2.70 | 2.93 | 2.87 | 1.63 | 1.97 | 0.10 | 1.33 |

**(B)**

**Model I** · **Model II** · **Model III**

**(C)**

**(D)**

**Fig. 3 | Optimization of Bˣʸ. A** LUMO energy level [eV], $\Delta G_w^\circ$ [kcal mol⁻¹], and $\Delta G_H^\ddagger$ [kcal mol⁻¹] for selected boranes **Bˣʸ**. TOFs [h⁻¹] calculated based on the yield of **3aa** under the model conditions are also shown. **B** Gaussian process regression using the programming library GPy for the prediction of the TOF values. The theoretical values of the parameters $\Delta G_w^\circ$ and $\Delta G_H^\ddagger$ (**Model I**), the LUMO level and $\Delta G_H^\ddagger$ (**Model II**), or the LUMO level and $\Delta G_w^\circ$ (**Model III**) were used. **C** Comparison of the TOF values predicted using **Models I, II**, or **III**. Error bars represent 1σ standard deviation. **D** Comparison of the experimental and predicted TOF values for **B⁴ᵇ**, **B⁴ᶜ**, and **B⁴ᵉ**.

frequency related to the H−H bond cleavage; however, to reduce the calculation costs, we performed an IRC calculation only for selected cases and confirmed their validity as a possible transition state structure.

Next, we turned our attention to collecting experimental data for the reported triarylboranes (**Bˣᵃ**, *x* = **1−3, 5, 6**; **B¹ᵛ**; **B¹ʷ**; B(C₆F₅)₃) and newly synthesized boranes (**B¹ᶠ**; **B²ʸ**, *y* = **b, c, e**; **B³ʸ**, *y* = **b, c, s**); the latter compounds were used to analyze the influence of the derivatization of

the 2,6-$Cl_2$-$C_6H_3$ structure. We obtained the turnover frequency (TOF in $h^{-1}$; Fig. 3A) as an experimental parameter calculated based on the yield of **3aa** from the reductive alkylation of amino acid **1a** with benzaldehyde (**2a**) and $H_2$ in the presence of 5 mol% **B$^{xy}$** under the shown conditions as a model reaction (Fig. 2B). The design of this model reaction is based on the potential of **B$^{1a}$** to produce **3aa** in approximately 50% yield after 6 h (i.e., TOF = 1.67 $h^{-1}$), as this consideration allows for a clearer evaluation of the positive and negative effects of structural derivatization during the optimization of **B$^{xy}$**.

### Reaction conditions optimization

With the theoretical and experimental parameters in hand, Gaussian process regression (GPR), which is a radial basis function kernel-based statistical-learning algorithm, using GPy (a programming library for GPR)[51], was applied to construct a model to predict the TOF values for the production of **3aa** (it should be noted here that unless stated otherwise, mean values are presented for the theoretically predicted TOF values). GPR using GPy constructs a regression model using a limited number of observed data through ML and searches for a subsequent adequate parameter value of **B$^{xy}$** using the surrogate model. Note that GPR analysis has been widely explored in depth in information science and is characterized by the availability of a wide range of acquisition functions. Hence, its performance and flexibility to address various types of problems can be expected to efficiently promote the exploration of wide-ranging experimental conditions[52]. We evaluated the accuracy of each GPR model based on the coefficient of determination ($Q^2$) of the leave-one-out (LOO) cross-validation using either training or validation data. It is noteworthy that higher $Q^2$ values approaching 1 indicate a more robust explanation for the data. We carried out the initial GPR analysis using the experimental TOF values and pairs of two of the three theoretical parameters (LUMO energy level, $\Delta G_w°$, and $\Delta G_H^‡$) obtained for the 15 boranes shown in Fig. 3A as training data. This GPR analysis resulted in three distinct models, labelled **Model I** ($\Delta G_w°$ vs $\Delta G_H^‡$; $Q^2 = 0.75$), **II** (LUMO level vs $\Delta G_H^‡$; $Q^2 = 0.76$), and **III** (LUMO level vs $\Delta G_w°$; $Q^2 = 0.21$) (Fig. 3B). A significantly lower $Q^2$ of **Model III** would indicate its insufficient reliability. Subsequently, we

used the theoretical parameters of the other 39 triarylboranes to predict their TOF values using these three models. We found intriguing inconsistencies among the TOF values for the **B$^4$** derivatives that contain *meta*-$CF_3$ groups predicted using **Model I** and those predicted using **Model II** (Fig. 3C), albeit that the $Q^2$ values of **Model I** and **II** were identical at this stage. For example, when **Model I** was applied, the TOF values of **B$^{4b}$**, **B$^{4c}$**, and **B$^{4e}$** were predicted to be 3.50, 3.66, and 3.59, respectively; however, using **Model II**, the corresponding TOF values were predicted to be 0.55, 0.23, and 0.55 respectively (Fig. 3C, D). A critical difference between **Model I** and **Model II** is the use of the LUMO energy levels in the GPR analysis. Thus, to evaluate whether the LUMO levels can serve here as a critical parameter for the prediction of the TOF for the production of **3aa**, we additionally synthesized **B$^{4b}$**, **B$^{4c}$**, and **B$^{4e}$**, and confirmed that these boranes demonstrate excellent activity for the reductive alkylation of **1a** with **2a** under the model reaction conditions as predicted by **Model I** (Fig. 3D). We updated each model additionally using the experimental TOF values of **B$^{4b}$**, **B$^{4c}$**, and **B$^{4e}$** to construct **Model I′** ($Q^2 = 0.73$), **II′** ($Q^2 = 0.28$), and **III′** ($Q^2 = 0.28$) (Supplementary Table 3). These results convinced us of the superiority of **Model I′** for the prediction of the catalytic activity of **B$^{xy}$** under the applied conditions. Based on synthetic accessibility considerations, we decided to employ **B$^{4b}$** as the optimal catalyst in the following experiments. These results also suggest that the employment of a combination of theoretical parameters related to the rate-determining step (e.g., $\Delta G_H^‡$ in this work) and the (potential) catalyst deactivation step (e.g., $\Delta G_w$ in this work) is decisive, whereas a parameter related to intrinsic Lewis acidity of triarylboranes, such as the LUMO energy level, is inappropriate for the prediction of the catalytic activity of triarylboranes under the applied conditions.

We also aimed to investigate relatively unexplored theoretical parameters for the construction of a regression-based model for the prediction of the catalytic activity of triarylboranes. In this context, we explored the deformation energy ($E_{DEF}$) [kcal $mol^{-1}$] that can be used to evaluate the degree of remote back-strain[49], where $E_{DEF}$ represents the energetic penalty associated with the change in the conformation at the boron center from trigonal planar to tetrahedral upon the

**(A)**

| | B$^{1a}$ | B$^{1f}$ | B$^{1v}$ | B$^{1w}$ | B$^{2a}$ | B$^{2b}$ | B$^{2c}$ | B$^{2e}$ | B$^{3a}$ | B$^{3b}$ | B$^{3c}$ | B$^{3s}$ | B$^{4b}$ | B$^{4c}$ | B$^{4e}$ | B$^{5a}$ | B$^{6a}$ | B($C_6F_5$)$_3$ |
|---|---|---|---|---|---|---|---|---|---|---|---|---|---|---|---|---|---|---|
| $E_{DEF}(H_2O)$ | +21.6 | +25.4 | +17.6 | +16.2 | +20.6 | +22.1 | +22.0 | +23.6 | +20.0 | +21.7 | +21.7 | +21.4 | +21.9 | +23.3 | +22.4 | +21.0 | +20.3 | +18.4 |
| $E_{DEF}(THF)$ | +27.7 | +31.4 | +27.8 | +23.0 | +28.6 | +29.9 | +30.1 | +29.8 | +28.5 | +29.9 | +30.1 | +31.2 | +29.7 | +29.9 | +29.9 | +30.9 | +31.2 | +25.8 |
| $q$(B) | +0.31 | +0.31 | +0.15 | +0.25 | +0.27 | +0.30 | +0.33 | +0.27 | +0.32 | +0.28 | +0.28 | +0.26 | +0.27 | +0.26 | +0.37 | +0.24 | +0.26 | +0.37 |
| $q$(C) | −0.17 | −0.17 | −0.14 | −0.16 | −0.14 | −0.19 | −0.20 | −0.17 | −0.16 | −0.17 | −0.16 | −0.16 | −0.16 | −0.15 | −0.17 | −0.13 | −0.15 | −0.21 |

**(B)**

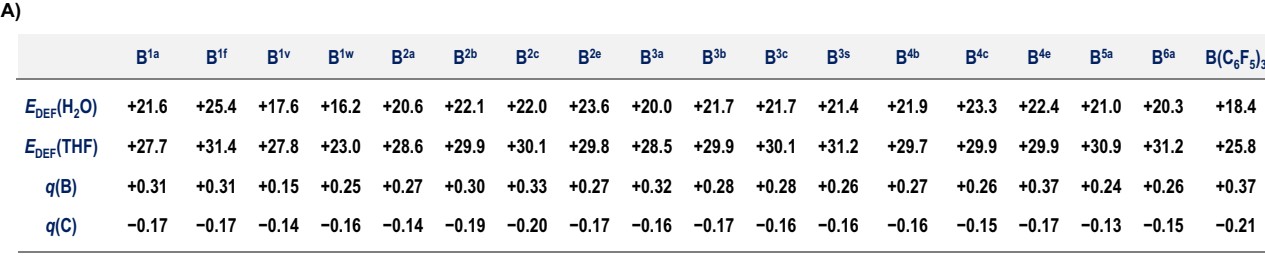
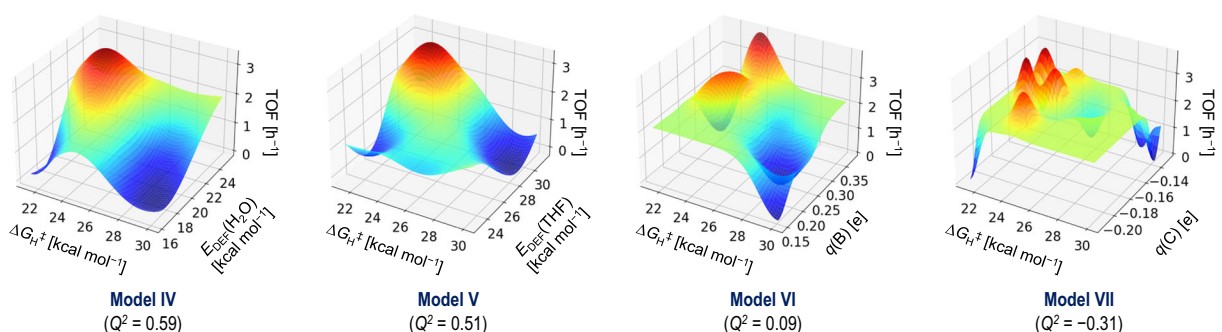

| Model IV | Model V | Model VI | Model VII |
|---|---|---|---|
| ($Q^2 = 0.59$) | ($Q^2 = 0.51$) | ($Q^2 = 0.09$) | ($Q^2 = -0.31$) |

**Fig. 4 | Exploring theoretical parameters. A** Theoretical parameters calculated for the selected **B$^{xy}$**. Deformation energies [kcal $mol^{-1}$], including $E_{DEF}(H_2O)$ and $E_{DEF}(THF)$, were calculated at the RI-DSD-PBEP86-D3BJ/ma-Def2-QZVPP//PBEh-3c/Def2-SVP level. Mulliken charges [e], $q$(B) (values on the boron atoms) and $q$(C)

(averaged values of three *ipso*-carbons), were calculated at the PBEh-3c/Def2-SVP level. **B** Gaussian process regression with GPy for the prediction of TOF values [$h^{-1}$], using $E_{DEF}(H_2O)$ and $\Delta G_H^‡$ (**Model IV**), $E_{DEF}(THF)$ and $\Delta G_H^‡$ (**Model V**), $q$(B) and $\Delta G_H^‡$ (**Model VI**), and $q$(C) and $\Delta G_H^‡$ (**Model VII**).

formation of adducts between Lewis bases (LBs) and triarylboranes[48,53,54]. We calculated the $E_{DEF}(LB)$ values for 18 boranes shown in Fig. 4A via the gas-phase optimization of their $H_2O$ or THF adducts (i.e., $E_{DEF}(H_2O)$ and $E_{DEF}(THF)$) followed by energy-decomposition analysis at the RI-DSD-PBEP86-D3BJ/ma-Def2-QZVPP//PBEh-3c/Def2-SVP level. The GPR analysis with $E_{DEF}(LB)$ and $\Delta G_H^{\ddagger}$ using the experimental TOF values resulted in the construction of **Model IV** (LB = $H_2O$; $Q^2$ = 0.59) and **V** (LB = THF; $Q^2$ = 0.51) with an acceptable reliability (Fig. 4B). Interestingly, these $Q^2$ values were found to be insensitive to the differences in $H_2O$ and THF. For further comparison, we also prepared **Model VI** ($q(B)$ vs $\Delta G_H^{\ddagger}$; $Q^2$ = 0.09) and **VII** ($q(C)$ vs $\Delta G_H^{\ddagger}$; $Q^2$ = –0.31), wherein Mulliken charges [e] on the boron atoms and their average values on three *ipso*-carbon atoms in **B**$^{xy}$ are given as $q(B)$ and $q(C)$, respectively; however, the reliability of these models was found to be insufficient. These results thus suggest that $E_{DEF}$ should be a valuable parameter to explore during the ML-based optimization of triarylboranes, as the estimation of $E_{DEF}$ through the optimization of structurally simple molecules (i.e., boranes, Lewis bases such as $H_2O$, and their adducts) is technically easier than the calculation of the activation energies, which is only feasible after the optimization of transition states.

With the optimal triarylborane **B**$^{4b}$ in hand, we modified the reaction conditions to reduce its environmental impact and thus establish a greener and more sustainable system. In this context, the use of an alternative reaction solvent that could also act as a Lewis base to generate an FLP with **B**$^{4b}$ was initially explored, given the recent demand for the replacement of hazardous THF with alternative ethereal compounds that exhibit lower toxicity combined with high chemical and thermal stability[32]. In the presence of 40 atm $H_2$, the reductive alkylation of **1a** with **2a** was carried out using THF, 2-methyltetrahydrofuran (2-MeTHF), cyclopentyl methyl ether (CPME), 4-methyltetrahydropyran (MTHP), or 2,2,4-trimethyl-1,3-dioxolane (TMD) (Fig. 5A). While THF provided a superior result (50%) compared to 2-MeTHF (38%) and CPME (18%), **3aa** was generated in 59% yield when MTHP was used. Prolongation of the reaction time to 24 h resulted in the formation of **3aa** in 72% yield; however, the removal of the 4 Å MS caused a decrease in the yield of **3aa** to 42%. Finally, increasing the $H_2$ pressure to 60 atm resulted in the formation of **3aa** in 95% yield after the period of 24 h. The use of TMD did not furnish any **3aa**. It should be noted that MTHP can be easily separated from water (its solubility in $H_2O$ is ~1.5 wt%) and removed under reduced pressure due to its strong hydrophobicity and low heat of vaporization, although its employment as a greener solvent has been limited in organic synthesis compared with the use of 2-MeTHF and CPME[32,33]. Moreover, to clarify the benefit of using MTHP over THF, we compared the activation energies for the heterolytic cleavage of $H_2$ by the combination of **B**$^{4b}$ and THF or MTHP at the $\omega$B97X-D/6-311+G(d,p)//$\omega$B97X-D/6-31G(d,p) level (Fig. 5B). A possible transition state was found in both cases, and that in the case of MTHP was found to be more stabilized ($TS_{MTHP}$ = +21.8 kcal mol$^{-1}$) than that in the case involving THF ($TS_{THF}$ = +23.2 kcal mol$^{-1}$). We attribute this stabilization to the increased structural flexibility of the tetrahydropyrane motif relative to THF, which allows the formation of efficient non-covalent interactions (NCIs) between the F/Cl atoms in **B**$^{4b}$ and the H atoms in MTHP. The participation of such NCIs was confirmed using the quantum theory of atoms in molecules (AIM) method (for details, see Supplementary Fig. 13)[55,56].

## Functional-group compatibility

The **B**$^{4b}$-catalyzed reductive alkylation of **1a** with **2a** in MTHP using $H_2$ (40 atm) demonstrated remarkable compatibility toward a variety of additives (**A0**–**A21**) (Fig. 6). All these experiments were carried out twice, and mean values [%] are given for the yield of **3aa**, the recovered additive, and the imine intermediate 4-(benzylideneamino)benzoic acid formed in situ through the **B**$^{4b}$-catalyzed condensation of **1a** and **2a** (left

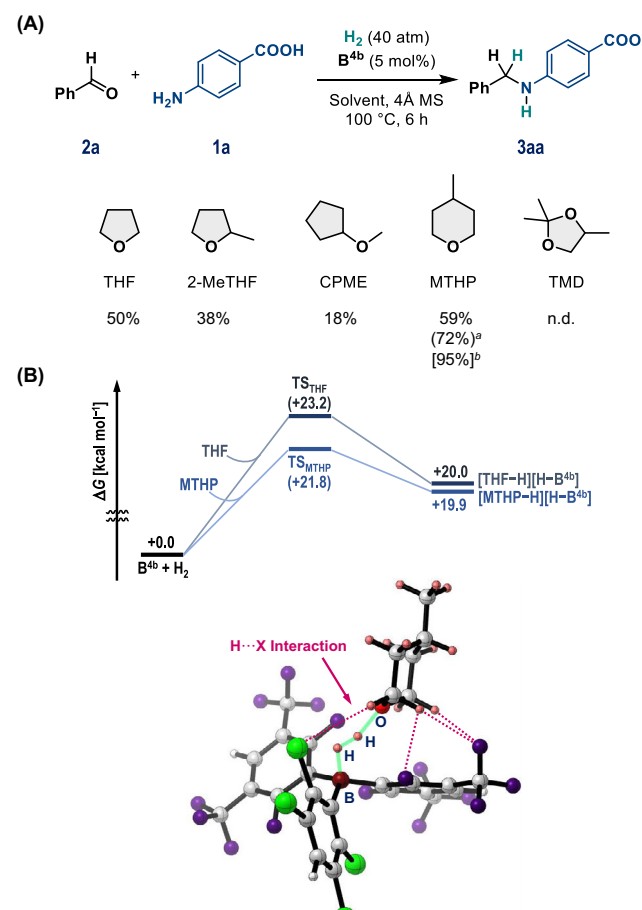

**(A)**

THF (40 atm)
**B**$^{4b}$ (5 mol%)
Solvent, 4Å MS
100 °C, 6 h

| THF | 2-MeTHF | CPME | MTHP | TMD |
|-----|---------|------|------|-----|
| 50% | 38% | 18% | 59% (72%)$^a$ [95%]$^b$ | n.d. |

**(B)**

$TS_{THF}$ (+23.2)
$TS_{MTHP}$ (+21.8)
THF
MTHP
+20.0 [THF-H][H-B$^{4b}$]
+19.9 [MTHP-H][H-B$^{4b}$]
+0.0 **B**$^{4b}$ + $H_2$
$\Delta G$ [kcal mol$^{-1}$]

H···X Interaction

**Fig. 5 | Optimization of reaction conditions. A** Exploration of greener Lewis-basic solvents. Reaction conditions: **1a** (0.4 mmol, 0.05 M), **2a** (1.0 equiv.), **B**$^{4b}$ (5 mol%), and 4 Å MS (100 mg) were mixed in the solvent, followed by pressurization with $H_2$ (40 atm). The yield of **3aa** was determined via $^1$H NMR analysis. $^a$24 h. $^b$60 atm $H_2$ for 24 h. **B** Relative Gibbs free energies [kcal mol$^{-1}$] with respect to [**B**$^{4b}$ + $H_2$ + LB], where LB is either MTHP or THF, calculated at the $\omega$B97X-D/6-311+G(d,p)//$\omega$B97X-D/6-31G(d,p) level. The structure of $TS_{MTHP}$ is also shown. Pairs of atoms involved in H···X (F/Cl) interactions that were found using AIM analysis are indicated by dashed lines (H: pink; B: brown; C: gray; O: red; F: purple; Cl: light green).

cycle in Fig. 1B). Initially, we carried out a control experiment using **A0**, which confirmed that the yields of **3aa** and the remaining imine were consistent (72% and 28%, respectively) with those of the reaction conducted without **A0** (Figs. 5A and 6). Relative to the control experiment, the reductive alkylation among **1a, 2a**, and $H_2$ proceeded without a significant change in the yield of **3aa**, the recovered additives, or imine intermediates for additives with ketone (**A3/A13**), primary amide (**A4**), aryl bromide/iodide including alkyl ether (**A6/A7**), allylic ether (**A8**), terminal alkyne (**A9**), enone (**A12**), nitrile (**A15**), and ester (**A20**) moieties. It is also noteworthy that additives including sulfhydryl (**A16**) and sulfide (**A17**) moieties did not affect the present reaction, whereas such sulfur-containing compounds can be critical inhibitors in transition-metal-based catalysis and organocatalysis[42]. On the other hand, the hydrogenation of the imine intermediates was suppressed in the presence of aliphatic/aromatic carboxy (**A1/A14**), aliphatic hydroxyl (**A2**), bulky silyl ether (**A10**), pinacolatoboryl (**A18**), and *N-tert*-butoxycarbonyl (Boc) (**A21**) moieties, as these functional groups include either a Lewis-basic or -acidic site that can kinetically inhibit the formation of the FLP consisting of **B**$^{4b}$ and MTHP. In fact, the quantitative recovery of the additives after a period of 24 h was confirmed, without the generation of any other significant byproduct, i.e., the sum of the yields of **3aa** and the imine was always ~95%. In contrast, the formation of **3aa** was largely

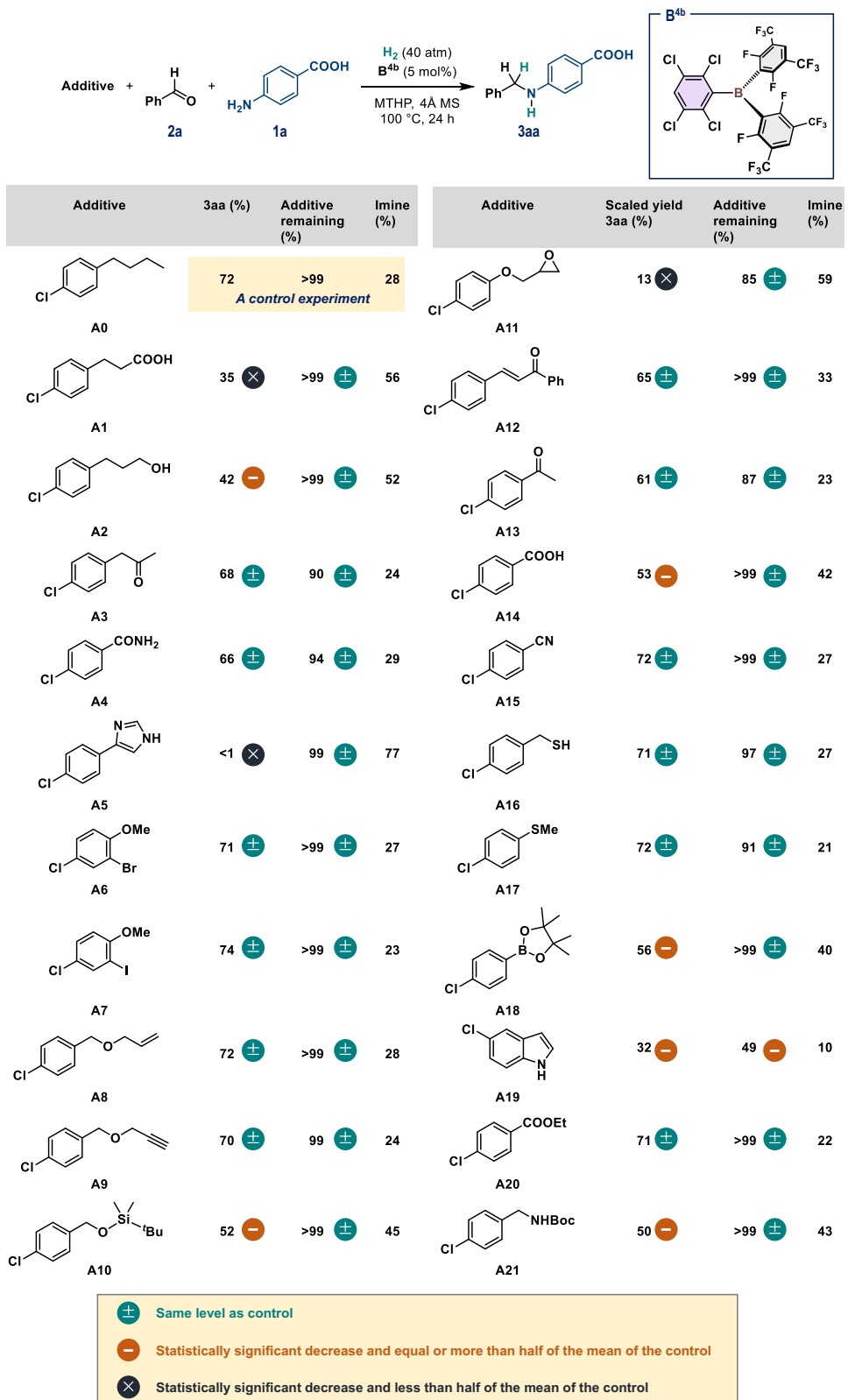

**Fig. 6 | Exploring functional-group compatibility.** Reaction conditions: **1a** (0.4 mmol, 0.05 M), **2a** (1.0 equiv.), additive (1.0 equiv.), **B**[4b] (5 mol%), and 4 Å MS (100 mg) were mixed in MTHP, followed by pressurization with $H_2$ (40 atm). The mean values of two experiments are given, which were determined via $^1$H NMR analysis.

suppressed under reaction conditions including *N*-heteroaromatic moieties such as an imidazole (**A5**) and an indole (**A19**), as these heteroaromatic units can react with **B**[4b] to form Lewis adducts and/or with aldehydes to complicate the system. In the case of **A11**, which includes an epoxide moiety, **3aa** was only produced in 13% yield, and a significant

loss of **A11** was confirmed after the reaction. Given that Lewis-acidic triarylboranes mediate the ring-opening transformation of epoxides[57,58], the **B**[4b]-catalyzed ring-opening reaction of **A11** to give the corresponding aldehyde can be expected to compete with the targeted reaction (see Supplementary Information for details).

## Scope study

Finally, we applied the combination of **B[4b]** and MTHP for the reductive alkylation of aniline-derived amino acids and peptide derivatives in the presence of H$_2$ (Fig. 7). Aminosalicylic acids **1b** and **1c** were effectively alkylated under the optimized conditions, and **3ba** and **3ca** were obtained in 90% and 93% yield, respectively; 60 atm of H$_2$ was used in the latter case. For comparison, under a pressure of 80 atm of H$_2$ in THF, **3ba** and **3ca** were furnished in 47% and 70% yield in the presence

**Fig. 7 | The B[4b]-catalyzed reductive alkylation of aniline derivatives (1b–1s) with aldehydes (2a–d) using H$_2$ in MTHP.** General conditions: **1** (0.4 mmol, 0.05 M), **2** (1.0 equiv.), **B[4b]** (5 mol%), and 4 Å MS (100 mg) were mixed in MTHP, followed by pressurization with H$_2$. Yields of isolated products are given. [a]60 atm H$_2$. [b]The formation of the imine in >99% was confirmed.

of 10 mol% and 15 mol% **B$^{1a}$**, respectively, which again demonstrates the advantages of the present system using **B$^{4b}$** and MTHP in terms of synthetic efficiency and sustainability. The reductive alkylation of anthranilic acid (**1d**), which is also known as vitamin L1, 5-aminoisophthalic acid (**1e**), and 4-aminophenylacetic acid (**1f**) afforded **3da**, **3ea**, and **3fa** in excellent yield using 40–60 atm of H$_2$. In contrast, we recognized that aliphatic amino acids (or their imine derivatives) and substrates insoluble in MTHP were not suitable. For example, aspartic acid (**1g**) is insoluble in MTHP, and no reaction took place when **1g** was employed under otherwise identical conditions. Esterification of the carboxy group in 3-amino-4,4-dimethylpentanoic acid effectively improved its solubility in MTHP; however, the hydrogenation of the imine derived from **1h** and **2a** did not occur. Based on these results, we prepared alanine- and S-methylcysteine-based peptides **1i** and **1j**, and subjected them to the optimal reaction conditions; alkylated peptides **3ia** and **3ja** were obtained in 95% and 55% yield, respectively. In terms of the scope of aldehydes, 3,5-bis(trifluoromethyl)benzaldehyde (**2b**) furnished **3ab** in 93% yield, but 3,5-di-tert-butylbenzaldehyde (**2c**) gave **3ac** in merely 38%. In the latter case, a significant amount of **2c** remained unreacted, indicating difficulties associated with the formation of the imine intermediate due to the decreased electrophilicity of the aldehyde moiety. A comparable result was obtained when p-tolualdehyde (**2d**) was used with respect to the case using **2a**, and **3ad** was afforded in 80% yield. As confirmed by the aforementioned robustness screening, the **B$^{4b}$**/MTHP system exhibited remarkable functional-group compatibility in the reductive alkylation of aniline derivatives to afford **3ka–3sa** in excellent yield. Given that harsh conditions, including 80 atm of H$_2$, 10 mol% borane, and/or a longer reaction time were required for the reactions with **1l, 1 m**, and **1q** in the reported **B$^{1a}$**/THF system, the present **B$^{4b}$**/MTHP system clearly demonstrates its advantages.

The present study demonstrates an in-silico-assisted approach to designing triarylboranes that exhibit promising reactivity as main-group catalysts for the reductive alkylation of aniline-derived amino acids with H$_2$. We have constructed an in-silico library of triarylboranes and obtained their theoretical parameters using DFT calculations. Guided by Gaussian process regression (GPR) using theoretical and experimental parameters, we identified the optimal triarylborane, i.e., B(2,3,5,6-Cl$_4$-C$_6$H)(2,6-F$_2$-3,5-(CF$_3$)$_2$-C$_6$H)$_2$ (**B$^{4b}$**). Through the evaluation of the regression-based models, we confirmed that the use of a parameter related to an intrinsic Lewis acidity of the triarylboranes (e.g., LUMO energy level and Mulliken charge on the boron atom) as one of the variables for the GPR analysis may lead to an underestimation when predicting the catalyst activity (TOF in h$^{-1}$) under the optimized reaction conditions. Moreover, we propose that the deformation energy ($E_{DEF}$) may serve as a potentially useful parameter to construct an adequate model. We also identified that 4-methyltetrahydropyran (MTHP) is a superior Lewis-basic solvent for not only the generation of FLP species with **B$^{4b}$**, but also for the realization of a more practical and less-hazardous reaction system compared to a system using THF. In fact, the **B$^{4b}$**-catalyzed reductive alkylation using aldehydes as an alkylating reagent and H$_2$ in MTHP proceeded efficiently even in the presence of a variety of additives, showcasing its broad functional-group compatibility. Aniline-derived amino acids and C-terminal-protected peptides were alkylated in good-to-excellent yields under the optimized conditions with the concomitant generation of H$_2$O as the sole byproduct.

## Methods

General procedures for reductive alkylation of **1x′** with **2y′** affording **3x′y′**: A 30 mL autoclave was charged with **1x′** (0.40 mmol), **2 y′** (0.40 mmol), **B$^{4b}$** (0.02 mmol), 4 Å MS (100 mg), and MTHP (8 mL). Once sealed, the vessel was pressurized with H$_2$ (40 or 60 atm), and the reaction mixture was stirred at 100 °C for 24 h. Then, degassed at rt followed by the addition of acetone, the resultant mixture was filtered to remove MS and other solids when generated. Subsequently, all volatiles were removed in vacuo to give **3x′y′**, which was purified by flash column chromatography on silica gel.

## Data availability

Data generated or analyzed during this study are provided in full within the published article and its supplementary materials. Metrical data for the solid-state structures are available from the Cambridge Crystallographic Data Centre (CCDC) under reference numbers 2295627 (**B$^{1f}$**), 2295628 (**B$^{2b}$**), 2295633 (**B$^{2c}$**), 2295634 (**B$^{2e}$**), 2295629 (**B$^{3b}$**), 2295635 (**B$^{3c}$**), 2295631 (**B$^{3s}$**), 2295632 (**B$^{4b}$**), and 2295630 (**B$^{4e}$**). These data can be obtained free of charge from the CCDC via www.ccdc.cam.ac.uk/data_request/cif. Coordinates of the optimized structures are provided as source data. All other data are available from the corresponding author. Source data are provided with this paper.

## Code availability

The code used in this work can be found in the Zenodo repository at https://doi.org/10.5281/zenodo.8420294.

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

## Acknowledgements

We thank Kuraray Co., Ltd., for providing 4-methyltetrahydropyran. This project was supported by Grants-in-Aid for Transformative Research Area (A) Digitalization-Driven Transformative Organic Synthesis (JSPS

KAKENHI grants 22H05363 to Y.Ho. and 21H05217 to S.T.) as well as the Environment Research and Technology Development Fund (JPMEERF20211R01 to Y.Ho.) of the Environmental Restoration and Conservation Agency of the Ministry of the Environment of Japan. Part of this work was supported by JST SPRING (grant JPMJSP2138 to Y.Hi.). Parts of the calculations were performed using resources from the Research Center for Computational Science, Okazaki, Japan (22-IMS-C107 and 23-IMS-C094).

## Author contributions

Y.Ho. conceived and directed this project. Y.Hi. and Y.Ho. performed the experiments and theoretical calculations. Y.Hi., T.W., and S.T. performed the Gaussian process regression. Y.Hi. prepared an initial draft of the manuscript, which was finalized by Y.Ho. with feedback from T.W., S.T., and S.O.

## Competing interests

The authors declare no competing interests.
