## [Peer Review File · Nature Communications]

REVIEWER COMMENTS

Reviewer #1 (Remarks to the Author):

The present study presents an ML-assisted approach for the catalytic reductive alkylation of amino acids with aldehydes and H₂. They constructed an in-silico library of triarylboranes, and optimized the gas-phase structures of these boranes using DFT calculations. The calculation methods employed are reasonable, and the obtained results are reliable. This work is of great significance to the field of catalytic chemistry. However, there are a few issues that need to be addressed as follows:

- 1) In the first sentence of the Introduction, a preposition is missing (Catalysis is a fact our daily lives).
- 2) On page 8, the author suggests that higher values of Q² indicate a more robust explanation for the data. However, it should be noted that both Model I and II have similar Q² values (0.75 vs 0.76). Moreover, intriguing inconsistencies were found among the TOF values predicted using Model I and II. It would be helpful if further clarification could be provided regarding this inconsistency.
- 3) The author only uses three theoretical parameters of boranes as training data but does not consider other calculation results such as the charge on the boron atom in borane. It would enhance the comprehensiveness of their analysis if additional relevant parameters were included.
- 4) There seems to be a discrepancy between the calculational methods used for gas-phase optimization of H₂O adduct (page 9) compared to those used for optimizing gas-phase structures of all 54 boranes (page 4). An explanation or justification for this difference would improve clarity.
- 5) The author conducted energy-decomposition analysis and chose deformation energy as training data; however, it may also be beneficial to consider other energies such as interaction energy.

Reviewer #2 (Remarks to the Author):

The manuscript by Hisata et al. describes the catalytic reductive functionalization of amino acids with H₂, using ML to guide decisions. The manuscript is well written and cites relevant work. Although the work is well executed from a wet laboratory point of view, I do have reservations on the actual utility of ML in this use case and the overall soundness of the approach. My comments are mostly related to ML, my field of expertise, and their order follows the order of appearance in the manuscript:

1. "(...) with H₂ through the construction of an in-silico library that includes a variety of unprecedented triarylboranes". What is the rationale for the library design, i.e. how were building blocks pooled? What expert knowledge is needed, or could this be automated with ML?
2. "54 triarylboranes, including 46 unprecedented structures". It is still not clear how big is the search space even after reading the manuscript a few times.
3. It is not clear the rationale for the THF substitute(s). This comment also applies to the statement in page 10. How did this choice bias the ML performance?
4. From a reader perspective it is not clear at all how do models I-III differ, how is uncertainty accounted for, if analyses were bootstrapped with different random states? How would these influence the outcome?
5. It is also not clear why the authors use GPR. Were different methods assessed and GPR emerged as the most promising for the use case in hand?
6. LOO cross-validation (CV) is far too optimistic and should never be used to gauge the model predictiveness or generalizability. The authors should consider switching to 5 or 10-fold CV and MAE/MSE as metrics.
7. The accuracy of Model I (RMSE = 0.42 [h⁻¹]) was confirmed to be greater than that of Model II (RMSE = 2.78 [h⁻¹]) and Model III (RMSE = 1.22 [h⁻¹]). How would these values compare with the experimental error?
8. "Although we successfully developed Model I for the prediction of the catalyst activity of B_xy," This is a big statement, so it needs a thorough comparison, backed up by robust statistics. For example, what is the baseline method? This comment circles back to the others in 4) and 5).
9. "With the optimal triarylborane B4b in hand, we modified the reaction conditions to reduce its environmental impact and thus establish a greener and more sustainable system". How were these modified? Purely based on expert intuition or any hints from in silico studies?
10. "Aminosalicylic acids 1b and 1c were effectively alkylated under the optimized conditions, and 3ba and 3ca were obtained in 90% and 93% yield, respectively; 60 atm of H₂ was used in the latter case." This is an interesting result, but what was the role of ML to get here? This is the part I am most skeptical about, together with the general use of the approach in other scenarios.
11. "Through the evaluation of the regression-based models, we confirmed that the deformation energy (EDEF) may serve as a potentially useful parameter to construct an adequate model, while the use of LUMO energy levels tends to lead to an underestimation when predicting the catalyst activity (TOF in h⁻¹) under the model reaction conditions". How was the feature importance extracted?
12. The authors should consider making their data FAIR.

Overall, I do think this is an interesting study from the wet lab point of view, but the use of ML is very fuzzy, and it reads like a decorator rather than an enabler for some of the findings. Further, the ML part is only briefly described, and one would not be able to reproduce the methods by reading the manuscript. Because I am evaluating the manuscript on the ML merits, I feel the novelty and/or relevance of the ML is not at the level expected from a manuscript in Nature Communications.

Reviewer #3 (Remarks to the Author):

The manuscript by Hoshimoto and coworkers elucidates a cheminformatics-based, machine learning-assisted approach for identifying triarylborane catalysts in the development of reductive aminations involving aldehydes and aniline derivatives in THF. An in-silico library of triarylboranes was constructed, and their theoretical parameters were determined through DFT calculations. Subsequent synthesis of new triarylboranes as Lewis-acid catalyst candidates, along with a logical iterative analysis of theoretical/experimental turnover frequencies (TOF) in the model reaction, revealed the optimal triarylborane, which contains a 2,3,5,6-tetrachloro-substituted phenyl ring, along with two other phenyl rings bearing fluoro and trifluoromethyl groups, and deformation energy emerged as a valuable parameter for constructing regression-based models. Furthermore, the authors discovered that 4-methyltetrahydropyran (MTHP) can serve as both a solvent and a Lewis basic component in FLP for H₂ cleavage in this reaction, replacing THF, which has toxicity concerns, supported by DFT calculations on the transition states. The manuscript also explores functional group tolerance by introducing various additives (Figure 6) and provides a brief scope of the reductive amination catalyzed by a triarylborane catalyst.

While acknowledging that the new FLP combinations (new triarylborane and MTHP) are not easily accessible or inexpensive, and the scope of the reaction is quite narrow, the manuscript showcases a commendable machine learning-assisted approach for catalyst identification. It presents a comprehensive comparison with experimental results, offering valuable insights into new catalyst development and their applications. This reviewer concludes that this work could significantly impact the readership of Nature Communications and recommends its publication after addressing the following comments.

(1) The manuscript could be significantly improved by including the catalytic cycle for this FLP-promoted reductive amination.

(2) The scope of the reaction is limited to aniline derivatives containing a carboxylic acid or an amide substituent, in contrast to the broader implication suggested by the title, which mentions 'the reductive functionalization of amino acids.' Please revise the title to accurately reflect the actual scope of the reaction.

(3) Although the manuscript doesn't primarily emphasize the novelty of the reaction, it would be advantageous to provide insights into why these particular catalysts are effective for reactions involving anilines with a carboxylic acid substituent. Additionally, it would be insightful to explore whether this catalyst is also effective for anilines without a carboxylic acid substituent.

Additions to and Corrections to the Manuscript

Responses to Referee 1

Thank you very much for your positive evaluation of our manuscript (“This work is of great significance to the field of catalytic chemistry”). We have revised the original manuscript based on your valuable and critical suggestions, which have undoubtedly made our manuscript more readable and stringent. Please kindly confirm the following answers and revisions in response to your comments.

Q1: In the first sentence of the Introduction, a preposition is missing (Catalysis is a fact our daily lives).

A1: We should have carefully checked this before submission. We have corrected the first sentence as follows and checked all parts of the manuscript carefully again.

Revision to the first sentence

“Catalysis is a fact **of** our daily lives.”

Q2: On page 8, the author suggests that higher values of Q^2 indicate a more robust explanation for the data. However, it should be noted that both Model I and II have similar Q^2 values (0.75 vs 0.76). Moreover, intriguing inconsistencies were found among the TOF values predicted using Model I and II. It would be helpful if further clarification could be provided regarding this inconsistency.

A2: Thank you for this suggestion. We initially compared the accuracy of our models (i.e., the Q^2 value) that were constructed by using 15 sets of training data. At this stage, **Model I** and **II** exhibited identical Q^2 values, but **Model III** showed significantly lower accuracy. These results suggest that the contribution of $\Delta G_{\text{H}}^\ddagger$ as one of the explanatory variables is crucial for the construction of a reliable model. Although **Model I** and **II** predict similar TOFs for most of the boranes in the library, we noticed that the predicted TOFs for the **B⁴** derivatives differ significantly between these models, suggesting that one of these models alone cannot accurately predict the activity of these **B⁴** derivatives. Thus, we synthesized three **B⁴** derivatives and confirmed that **Model I** is the superior model under the applied conditions. This was also confirmed by updating each model using 18 sets of training data, including experimental TOFs of **B⁴** derivatives (Model I: $Q^2 = 0.73$; Model II: $Q^2 = 0.28$).

As suggested, this point is very intriguing and of paramount importance for deciding the direction of our research. We thus revised the manuscript and related Supporting Information accordingly:

Additions to the revised manuscript

Line 20~, p8:

“We found intriguing inconsistencies among the TOF values for the **B**⁴ derivatives that contain *meta*-CF₃ groups predicted using **Model I** and those predicted using **Model II** (Figure 3C), albeit that the Q^2 values of **Model I** and **II** were identical at this stage.”

Line 30~, p8:

“We updated each model additionally using the experimental TOF values of **B**^{4b}, **B**^{4c}, and **B**^{4e} to construct **Model I'** ($Q^2 = 0.73$), **II'** ($Q^2 = 0.28$), and **III'** ($Q^2 = 0.28$) (Table S3). These results convinced us of the superiority of **Model I'** for the prediction of the catalytic activity of **B**^y under the applied conditions.”

Line 1~, p9:

“These results also suggest that the employment of a combination of theoretical parameters related to the rate-determining step (e.g., $\Delta G_{\text{H}^\ddagger}$ in this work) and the (potential) catalyst deactivation step (e.g., ΔG_{w} in this work) is decisive, whereas a parameter related to the intrinsic Lewis acidity of triarylboranes, such as the LUMO energy level, is inappropriate for the prediction of the catalytic activity of triarylboranes under the applied conditions.”

Q3: The author only uses three theoretical parameters of boranes as training data but does not consider other calculation results such as the charge on the boron atom in borane. It would enhance the comprehensiveness of their analysis if additional relevant parameters were included.

A3: Thank you very much for this important suggestion. We also collected the following theoretical parameters related to the triarylboranes:

- (1) Mulliken charges on the boron atom → **Model VI** in Figure 4
- (2) Averaged Mulliken charges on three *ipso*-carbon atoms → **Model VII** in Figure 4
- (3) Mayer's bonded valences on the boron atom → **Table S3** in the Supporting Information
- (4) Averaged Mayer's bonded valences on three *ipso*-carbon atoms → **Table S3** in the Supporting Information
- (5) Deformation energies obtained from borane-THF adducts ($E_{\text{DEF}}(\text{THF})$) → **Model IV** in Figure 4

The obtained values in (3) and (4) are almost identical for all boranes used in this study, and we therefore decided to only employ the parameters in (1), (2), and (5) for the GPR analysis. Thus, we were able to confirm that the employment of theoretical parameters related to the

intrinsic Lewis acidity of the boranes (i.e., LUMO levels and Mulliken charges on the boron and *ipso*-C atoms) tends to underestimate the catalytic activity of triarylboranes, probably because these parameters do not consider the geometrical change from trigonal planar to tetrahedral that must occur during their catalysis.

We also discussed the influence of the Lewis-basic partner involved in the calculation of E_{DEF} based on the comparison of $E_{\text{DEF}}(\text{H}_2\text{O})$ and $E_{\text{DEF}}(\text{THF})$. Interestingly, although $E_{\text{DEF}}(\text{H}_2\text{O})$ values tend to be larger than values in $E_{\text{DEF}}(\text{THF})$, the accuracy of the constructed regression models combined with $\Delta G_{\text{H}}^\ddagger$ is fully comparable (cf. Figure 4B below). These results suggest that theoretical parameters related to the deformation of the boranes can be useful for predicting their catalytic activity. Moreover, the influence of Lewis-basic partners should be limited; hence, one can use $E_{\text{DEF}}(\text{H}_2\text{O})$, which is far easier to calculate than larger Lewis bases.

Based on these considerations, we have revised Figure 4 and the corresponding discussion as follows:

Revision to the discussions related to Figure 4 (pp 9-10)

“We also aimed to investigate relatively unexplored theoretical parameters for the construction of a regression-based model for the prediction of the catalytic activity of triarylboranes. In this context, we explored the deformation energy (E_{DEF}) [kcal mol^{-1}] that can be used to evaluate the degree of remote back-strain,⁴⁹ where E_{DEF} represents the energetic penalty associated with the change in the conformation at the boron center from trigonal planar to tetrahedral upon the formation of adducts between Lewis bases (LBs) and homoleptic triarylboranes.^{48,53,54} We calculated the $E_{\text{DEF}}(\text{LB})$ values for 18 boranes shown in Figure 4A via the gas-phase optimization of their H_2O or THF adducts (i.e., $E_{\text{DEF}}(\text{H}_2\text{O})$ and $E_{\text{DEF}}(\text{THF})$) followed by energy-decomposition analysis at the RI-DSD-PBEP86-D3BJ/ma-Def2-QZVPP//PBEh-3c/Def2-SVP level. The GPR analysis with $E_{\text{DEF}}(\text{LB})$ and $\Delta G_{\text{H}}^\ddagger$ using the experimental TOF values resulted in the construction of **Model IV** (LB = H_2O ; $Q^2 = 0.59$) and **V** (LB = THF; $Q^2 = 0.51$) with an acceptable reliability (Figure 4B). Interestingly, these Q^2 values were found to be insensitive to the differences in H_2O and THF. For further comparison, we also prepared **Model VI** ($q(\text{B})$ vs $\Delta G_{\text{H}}^\ddagger$; $Q^2 = 0.09$) and **VII** ($q(\text{C})$ vs $\Delta G_{\text{H}}^\ddagger$; $Q^2 = -0.31$), wherein Mulliken charges, [e], on the boron atoms and their average values on three *ipso*-carbon atoms in \mathbf{B}^{xy} are given as $q(\text{B})$ and $q(\text{C})$, respectively; however, the reliability of these models was found to be insufficient. These results thus suggest that E_{DEF} should be a valuable parameter to explore during the ML-based optimization of triarylboranes, as the estimation of E_{DEF} through the optimization of structurally simple molecules (i.e., boranes, Lewis bases such as H_2O , and their adducts) is technically easier than the calculation of the

activation energies, which is only feasible after the optimization of transition states.”

Figure 4. (A) Theoretical parameters calculated for the selected \mathbf{B}^{xy} . Deformation energies [kcal mol⁻¹], including $E_{\text{DEF}}(\text{H}_2\text{O})$ and $E_{\text{DEF}}(\text{THF})$, were calculated at the RI-DSD-PBEP86-D3BJ/ma-Def2-QZVPP//PBEh-3c/Def2-SVP level. Mulliken charges, [e], $q(\text{B})$ (values on the boron atoms) and $q(\text{C})$ (averaged values of three *ipso*-carbons), were calculated at the PBEh-3c/Def2-SVP level. (B) Gaussian process regression with GPy for the prediction of TOF values [h⁻¹], using $E_{\text{DEF}}(\text{H}_2\text{O})$ and $\Delta G_{\text{H}}^{\ddagger}$ (**Model IV**), $E_{\text{DEF}}(\text{THF})$ and $\Delta G_{\text{H}}^{\ddagger}$ (**Model V**), $q(\text{B})$ and $\Delta G_{\text{H}}^{\ddagger}$ (**Model VI**), and $q(\text{C})$ and $\Delta G_{\text{H}}^{\ddagger}$ (**Model VII**).

Q4: There seems to be a discrepancy between the calculational methods used for gas-phase optimization of H₂O adduct (page 9) compared to those used for optimizing gas-phase structures of all 54 boranes (page 4). An explanation or justification for this difference would improve clarity.

A4: We have changed the employed theoretical conditions depending on the target parameters as shown below:

- (1) $\omega\text{B97X-D}/6\text{-}311\text{+G(d,p)}/\omega\text{B97X-D}/6\text{-}31\text{G(d,p)}$ level for the optimization of gas-phase structures and the calculation of LUMO levels, $\Delta G_{\text{w}}^{\circ}$, and $\Delta G_{\text{H}}^{\ddagger}$
- (2) RI-DSD-PBEP86-D3BJ/ma-Def2-QZVPP//PBEh-3c/Def2-SVP level for the calculation of $E_{\text{DEF}}(\text{H}_2\text{O})$ and $E_{\text{DEF}}(\text{THF})$
- (3) PBEh-3c/Def2-SVP level for the calculation of Mulliken charges and Mayer valences

In order to avoid any potential ambiguities regarding these conditions, we have revised the

manuscript as shown below (please also confirm the footnotes in Figure 4 shown on the previous page):

Additions to the revised manuscript

Line 25~, p5:

“Thus, we obtained the following energetic parameters, calculated at the ω B97X-D/6-311+G(d,p)// ω B97X-D/6-31G(d,p) level: (i) the energy levels of the LUMOs [eV], which include the p orbitals on the boron atoms~”

Q5: The author conducted energy-decomposition analysis and chose deformation energy as training data; however, it may also be beneficial to consider other energies such as interaction energy.

A5: Thank you very much for this constructive suggestion. In this work, we did not employ interaction energies, as we considered that such interaction energies (E_{INT}) should be covered by $\Delta G_{\text{w}}^{\circ}$ and $E_{\text{DEF}}(\text{H}_2\text{O})$, i.e., $E_{\text{INT}} = \Delta G_{\text{w}}^{\circ} + E_{\text{DEF}}(\text{H}_2\text{O})$. However, we agree that E_{INT} (related to the global Lewis acidity) and other energetical parameters (e.g., NCIs including the London dispersion and van-der-Waals interactions) can be explored as potential parameters for regression, which will be addressed in our future work.

Responses to the Referee 2

We deeply appreciate your suggestions and comments from the perspective of an expert in machine learning (ML) and your positive evaluation of our results (“I do think this is an interesting study from the wet-lab point of view”). We agree that our present approach is largely based on human intuition in order to compensate for a lack of experimental training data. As mentioned in the manuscript, chemo-informatics-assisted approaches in the field of Lewis-acid catalysis (especially triarylborane catalysis) have been significantly underdeveloped compared to the well-explored Lewis-basic (or Lewis-base-supported metal) catalysis. Thus, we have decided to explore this challenging task for the further development of the triarylborane catalysis, and we managed to eventually apply this strategy to generate the first examples of the reductive functionalization of peptides with H₂.

The critical suggestions of the three reviewers have certainly strengthened our manuscript, especially the ML parts, and accordingly, we would like to ask you to re-evaluate the manuscript.

Q6: “(...) with H₂ through the construction of an in-silico library that includes a variety of unprecedented triarylboranes”. What is the rationale for the library design, i.e. how were building blocks pooled? What expert knowledge is needed, or could this be automated with ML?

A6: Please kindly confirm the discussion for the construction of the *in-silico* borane library, given on p4. This discussion includes the following three points:

- (1) We generally explored triarylboranes that seemed to be synthetically accessible using common procedures. This is simply due to the considerable limitations associated with the synthesis of triarylboranes.
- (2) We have previously identified **B^{1a}** as a useful catalyst for the reductive functionalization of amines through classical trial-and-error experimental approaches. Thus, in this study, we have prepared more than 50 *in-silico* derivatives of **B^{1a}**. For these derivatizations, we chose the introduction of F, Cl, Br, CF₃, and aryl groups based on our extensive experience.
- (3) Based on detailed mechanistic studies, we understood two key species, i.e., a transition state in the rate-determining step and a resting state that triggers the irreversible decomposition of the triarylboranes. Thus, we focused on collecting the theoretical parameters related to these two species in this study.

Q7: “54 triarylboranes, including 46 unprecedented structures”. It is still not clear how big is the search space even after reading the manuscript a few times.

A7: Thank you for this question. So far, synthetic chemists have not sufficiently explored the

space of triarylboranes due to their limited availability in both experiments and theory. In fact, our previous work (*J. Am. Chem. Soc.* **2018**, *140*, 7292) only explored five boranes, while there are many reports that use only $\text{B}(\text{C}_6\text{F}_5)_3$ for the reaction developments. Against this background, this manuscript has significantly expanded the library of triarylboranes to 54 compounds, including 10 newly synthesized compounds. Nevertheless, we acknowledge that our present exploration is still limited. Thus, we are further expanding the library of triarylboranes to explore a bigger search space, which will be reported in the near future.

Q8: It is not clear the rationale for the THF substitute(s). This comment also applies to the statement in page 10. How did this choice bias the ML performance?

A8: Exploring the optimal Lewis-basic solvents was not involved in the ML processes. These substituents were predominantly optimized by considering the environmental impact to construct a greener and sustainable reaction system after the ML-assisted optimization of the borane catalyst.

Q9: From a reader perspective it is not clear at all how do models I-III differ, how is uncertainty accounted for, if analyses were bootstrapped with different random states? How would these influence the outcome?

A9: Thank you very much for this suggestion. Firstly, we have updated these models to **Model I'** ($Q^2 = 0.72$), **II'** ($Q^2 = 0.28$), and **III'** ($Q^2 = 0.28$) using 18 sets of training data in the revised manuscript (please kindly confirm also **A2** for details). Secondly, we have compared the averaged errors of 36 boranes that were not experimentally evaluated among these updated models and confirmed that **Model I'** (averaged error: 0.33) exhibits a higher reliability and smaller error than **Model II'** (averaged error: 0.40) or **III'** (averaged error: 0.63).

Q10: It is also not clear why the authors use GPR. Were different methods assessed and GPR emerged as the most promising for the use case in hand?

A10: We chose GPR in this study because GPR analysis has been widely explored in depth in information science and is characterized by the availability of a wide range of acquisition functions. Moreover, its performance and flexibility to address various types of problems can efficiently promote the exploration of the experimental conditions, as demonstrated by previous work of some of the co-authors (cf. Chem. Commun. 2020, 56, 1259–1262). Therefore, we have added the following sentence:

Additions to the revised manuscript

Line 7~, p8:

Department of Applied Chemistry, Faculty of Engineering, Suita, Osaka 565-0871, Japan
Phone +81-6-6879-7393, Fax +81-6-6879-7394
E-mail hoshimoto@chem.eng.osaka-u.ac.jp, ogoshi@chem.eng.osaka-u.ac.jp

“Note that GPR analysis has been widely explored in depth in information science and is characterized by the availability of a wide range of acquisition functions. Hence, its performance and flexibility to address various types of problems can be expected to efficiently promote the exploration of wide-ranging experimental conditions.⁵²”

Q11: LOO cross-validation (CV) is far too optimistic and should never be used to gauge the model predictiveness or generalizability. The authors should consider switching to 5 or 10-fold CV and MAE/MSE as metrics.

A11: As mentioned above, we only used 15-18 set of training data for the GPR analysis. With such a limited amount of data sets, LOO should be the optimal choice for CV.

Q12: The accuracy of Model I (RMSE = 0.42 [h⁻¹]) was confirmed to be greater than that of Model II (RMSE = 2.78 [h⁻¹]) and Model III (RMSE = 1.22 [h⁻¹]). How would these values compare with the experimental error?

A12: Thank you for this question. We generally determined experimental TOFs [h⁻¹] for 15 and 18 sets of training data only by a single experiment, and thus, it makes little sense to determine the experimental error. However, in the revised manuscript, we have discussed the corresponding accuracy of the regression models based on Q^2 . Please kindly confirm the discussions related to Figures 3 and 4 in the revised manuscript.

Q13: “Although we successfully developed Model I for the prediction of the catalyst activity of Bxy,” This is a big statement, so it needs a thorough comparison, backed up by robust statistics. For example, what is the baseline method? This comment circles back to the others in 4) and 5).

A13: We agree with this suggestion and have removed this statement in the revised manuscript. Moreover, we have revised the manuscript as discussed in **A3**.

Q14: “With the optimal triarylborane B4b in hand, we modified the reaction conditions to reduce its environmental impact and thus establish a greener and more sustainable system”. How were these modified? Purely based on expert intuition or any hints from in silico studies?

A14: As already explained in **A8**, the corresponding exploration of Lewis-basic solvents was carried out without the support of ML.

Q15: “Aminosalicylic acids 1b and 1c were effectively alkylated under the optimized conditions, and 3ba and 3ca were obtained in 90% and 93% yield, respectively; 60 atm of H₂ was used in the latter case.” This is an interesting result, but what was the role of ML to get here? This is the part I am most skeptical about, together with the general use of the approach in

other scenarios.

A15: In contemporary organic synthesis, the optimization of the reaction conditions (i.e., catalysts, solvent, additives, temperature, and reaction time, to name a few) is generally carried out using a specific combination of substrates; in this work: **1a** and **2a**. Unfortunately, the thus determined ‘optimal’ conditions often change when using different combinations of substrates due to e.g., presence of functional groups in the substrates.

Here, due to the confines of the project, the ML-assisted optimization of the catalyst (**B^{4b}**) and the experimentally assisted optimization of other reaction conditions (i.e., solvent, H₂ pressure, temperature, and reaction time) were carried out by employing only the combination of **1a** and **2a**. Thus, it can be expected that the optimal conditions may vary upon changing the substrate combination.

Q16: “Through the evaluation of the regression-based models, we confirmed that the deformation energy (EDEF) may serve as a potentially useful parameter to construct an adequate model, while the use of LUMO energy levels tends to lead to an underestimation when predicting the catalyst activity (TOF in h⁻¹) under the model reaction conditions”. How was the feature importance extracted?

A16: We have not investigated the feature importance for each parameter. We have searched for useful combinations of two explanatory variables using the GPR analysis and found that the combination of ΔG_{H^\ddagger} and ΔG_w is decisive, i.e., the construction of **Model I** (or **I'**). In addition, **Model II'** and **III'**, which include the LUMO as one of the variables failed to predict the TOFs of the **B⁴** species, which suggests that the use of the LUMO energy levels tends to lead to an underestimation.

Q17: The authors should consider making their data FAIR.

A17: Thank you very much for this comment. As explained above, we understood that our research still relies on human experience, expertise, and intuition. Please kindly confirm that we thus carefully mention “*In-Silico-Assisted* Derivatization of Triarylboranes” in the title and “ML-assisted approach” in the abstract and main text. Regardless, we have acknowledged the importance of such ML-assisted approaches because the identification of **B^{4b}** would most likely be very challenging based only on human experience and intuition. Moreover, we will attempt an unbiased purely ML-driven approach for the optimization of Lewis-acid catalysis in the future based on your suggestions above.

Responses to the Referee 3

Thank you very much for your evaluation of our manuscript as “this work could significantly impact the readership of Nature Communications”. Please kindly confirm the following answers and revisions, which will surely enhance the quality and readability of our manuscript.

Q18: The manuscript could be significantly improved by including the catalytic cycle for this FLP-promoted reductive amination.

A18: Thank you very much for this constructive suggestion. We have added Figure 1B and cited this figure when discussing the details of the reaction mechanisms in the revised manuscript. As requested, this change should make our manuscript more readable.

Addition to the revised manuscript

Figure 1. (A) Schematic representation of the concept of this study. (B) Proposed dual catalysis of triarylborane in the catalytic reductive alkylation of amines with aldehydes; **B** = triarylborane; **LB** = THF or MTHP.

Q19: The scope of the reaction is limited to aniline derivatives containing a carboxylic acid or an amide substituent, in contrast to the broader implication suggested by the title, which mentions 'the reductive functionalization of amino acids.' Please revise the title to accurately reflect the actual scope of the reaction.

A19: As requested, we have revised the title as shown below:

Revision of the title

“*In-Silico*-Assisted Derivatization of Triarylboranes for the Catalytic Reductive Functionalization of Aniline-Derived Amino Acids and Peptides with H₂”

Q20: Although the manuscript doesn't primarily emphasize the novelty of the reaction, it would be advantageous to provide insights into why these particular catalysts are effective for reactions involving anilines with a carboxylic acid substituent. Additionally, it would be insightful to explore whether this catalyst is also effective for anilines without a carboxylic acid substituent.

A20: Thank you very much for this critical point. We have added 9 examples of substituted anilines that were relatively difficult to transform for the previously reported **B**^{1a}/THF system and confirmed the nearly quantitative conversion to the alkylated amines even under reduced H₂ pressure and catalyst loading. These results clearly demonstrate that the **B**^{4b}/MTHP combination can cleave H₂ more rapidly than the reported **B**^{1a}/THF system, probably due to the enhanced Lewis acidity and the increased remote back strain. We have revised Figure 7 and the related discussions accordingly.

Addition to the revised manuscript

Line 28~, p14

“As confirmed by the aforementioned robustness screening, the **B**^{4b}/MTHP system exhibited remarkable functional-group compatibility in the reductive alkylation of aniline derivatives to afford **3ka**–**3sa** in excellent yield. Given that harsh conditions, including 80 atm of H₂, 10 mol% borane, and/or a longer reaction time were required for the reactions with **1m**, **1q**, and **1l** in the reported **B**^{1a}/THF system, the present **B**^{4b}/MTHP system clearly demonstrates its advantages.”

Figure 7. The **B^{4b}**-catalyzed reductive alkylation of aniline derivatives (**1a–1s**) with aldehydes (**2a–d**) using H₂ in MTHP. General conditions: **1** (0.4 mmol, 0.05 M), **2** (1.0 equiv.), **B^{4b}** (5 mol%), and 4Å MS (100 mg) were mixed in MTHP, followed by pressurization with H₂. Yields of isolated products are given. ^a60 atm H₂. ^bThe formation of the imine in >99% was confirmed

Responses to the office

- ✚ We have added the following statement in the Data-availability section:
 “Metrical data for the solid-state structures are available from the Cambridge Crystallographic Data Centre (CCDC) under reference numbers 2295627 (**B^{1f}**), 2295628 (**B^{2b}**), 2295633 (**B^{2c}**), 2295634 (**B^{2e}**), 2295629 (**B^{3b}**), 2295635 (**B^{3c}**), 2295631 (**B^{3s}**), 2295632 (**B^{4b}**), and 2295630 (**B^{4c}**). These data can be obtained free of charge from the CCDC via www.ccdc.cam.ac.uk/data_request/cif.”

- ✚ We have added the “Code Availability” section as follows:
 “The code used in this work can be found in the Zenodo repository at <https://zenodo.org/doi/10.5281/zenodo.8420294>.”

- ✚ We have revised the Abstract based on the revision of the manuscript:
 “Cheminformatics-based machine learning (ML) has been employed to determine optimal reaction conditions, including catalyst structures, in the field of synthetic chemistry. However, such ML-focused strategies have remained largely unexplored in the context of catalytic molecular transformations using Lewis-acidic main-group elements, probably due to the absence of a candidate library and effective guidelines (parameters) for the prediction of the activity of main-group elements. Here, the construction of a triarylborane library and its application to an ML-assisted approach for the catalytic reductive alkylation of aniline-derived amino acids and C-terminal-protected peptides with aldehydes and H₂ is reported. A combined theoretical and experimental approach identified the optimal borane, i.e., B(2,3,5,6-Cl₄-C₆H)(2,6-F₂-3,5-(CF₃)₂-C₆H)₂, which exhibits remarkable functional-group compatibility toward aniline derivatives including amino acids and peptides in the presence of 4-methyltetrahydropyran and H₂. The present catalytic system generates H₂O as the sole byproduct.”

- ✚ We have revised the Conclusion part based on the revision of the manuscript:
 “The present study demonstrates an *in-silico*-assisted approach to designing triarylboranes that exhibit promising reactivity as main-group catalysts for the reductive alkylation of aniline-derived amino acids with H₂. We have constructed an *in-silico* library of triarylboranes, including 46 unprecedented boranes, and obtained their theoretical parameters using DFT calculations. Guided by Gaussian process regression (GPR) using theoretical and experimental parameters, we identified the optimal triarylborane, i.e., B(2,3,5,6-Cl₄-C₆H)(2,6-F₂-3,5-(CF₃)₂-C₆H)₂ (**B^{4b}**). Through the evaluation of the regression-based models, we confirmed that the use of a parameter related to an intrinsic

Lewis acidity of the triarylboranes (e.g., LUMO energy level and Mulliken charge on the boron atom) as one of the variables for the GPR analysis may lead to an underestimation when predicting the catalyst activity (TOF in h^{-1}) under the optimized reaction conditions. Moreover, we propose that the deformation energy (E_{DEF}) may serve as a potentially useful parameter to construct an adequate model. We also identified that 4-methyltetrahydropyrene (MTHP) is a superior Lewis-basic solvent for not only the generation of FLP species with \mathbf{B}^{4b} , but also for the realization of a more practical and less-hazardous reaction system compared to a system using THF. In fact, the \mathbf{B}^{4b} -catalyzed reductive alkylation using aldehydes as an alkylating reagent and H_2 in MTHP proceeded efficiently even in the presence of a variety of additives, showcasing its broad functional-group compatibility. Aniline-derived amino acids and C-terminal-protected peptides were alkylated in good-to-excellent yields under the optimized conditions with the concomitant generation of H_2O as the sole byproduct.”

REVIEWERS' COMMENTS

Reviewer #1 (Remarks to the Author):

The authors have now made reasonable changes and is acceptable.

Reviewer #2 (Remarks to the Author):

I acknowledge the effort that the authors put in to answer all the comments and concerns raised. It is now clear that I misunderstood some parts of the manuscript, and the provided clarifications are enough in most instances. From the authors' replies I am still very much puzzled about one topic:

Question 7 about search space size: The authors mention that they expand it to 54 compounds in this study. This is extremely small and makes me question again the real utility of ML in this case. As previously mentioned by myself, the use of ML seems like a decorated rather than an enabler for discovery. Without a clear rationale for showing the benefits of ML I am very skeptical about the significance/impact of the work in this specific area. Can the authors clarify the cost of experimentation? If it is prohibitively expensive / time consuming (e.g. one/two weeks per data point), then I retract my comment and concede a point of utility. There might be other plausible motivations that I am unaware of but they have to be explicit in the manuscript, so that the ML audience can appreciate the work in full. Otherwise, the use of ML is not motivated, in my opinion.

Reviewer #3 (Remarks to the Author):

The revised manuscript by Hoshimoto and colleagues presents a cheminformatics-based, machine learning-assisted strategy for developing triarylborane catalytic systems for the reductive functionalization of aniline-derived amino acids and peptides. The authors have made revisions to address reviewers' comments, including revising the titles and adding the overall reaction

mechanism. Overall, this work is considered suitable for publication in Nature Communications after addressing the following minor point.

1. The catalytic cycle would benefit from the inclusion of charges for the chemical structures to enhance clarity.

Additions to and Corrections to the Manuscript

Responses to Referee 1

Thank you very much for your positive evaluation of our manuscript.

Responses to the Referee 2

Q1: Question 7 about search space size: The authors mention that they expand it to 54 compounds in this study. This is extremely small and makes me question again the real utility of ML in this case. As previously mentioned by myself, the use of ML seems like a decorated rather than an enabler for discovery. Without a clear rationale for showing the benefits of ML I am very skeptical about the significance/impact of the work in this specific area. Can the authors clarify the cost of experimentation? If it is prohibitively expensive / time consuming (e.g. one/two weeks per data point), then I retract my comment and concede a point of utility. There might be other plausible motivations that I am unaware of but they have to be explicit in the manuscript, so that the ML audience can appreciate the work in full. Otherwise, the use of ML is not motivated, in my opinion.

A1: Thank you very much for your time and suggestions. To clarify this point, we have added the following explanation in p.3:

Additions to the revised manuscript

“The synthesis of triarylboranes with unprecedented substitution patterns is typically a laborious and time-consuming process that often requires several weeks or even months for optimization. However, once the optimal procedures have been obtained, the optimized conditions can often be extrapolated, which is much faster. Therefore, using an ML-assisted approach to streamlining the selection of triarylborane candidates has the potential to significantly accelerate the optimization process and therefore the entire research process.”

Responses to the Referee 3

Q2: The catalytic cycle would benefit from the inclusion of charges for the chemical structures to enhance clarity.

A2: Thank you very much for this constructive suggestion. We have revised Figure 1B accordingly:

Addition to the revised manuscript

Department of Applied Chemistry, Faculty of Engineering, Suita, Osaka 565-0871, Japan
Phone +81-6-6879-7393, Fax +81-6-6879-7394
E-mail hoshimoto@chem.eng.osaka-u.ac.jp, ogoshi@chem.eng.osaka-u.ac.jp

Figure 1. (A) Schematic representation of the concept of this study. (B) Proposed dual catalysis of triarylborane in the catalytic reductive alkylation of amines with aldehydes; **B** = triarylborane; **LB** = THF or MTHP.